# Structural Insights into the Marine Alkaloid Discorhabdin G as a Scaffold towards New Acetylcholinesterase Inhibitors

**DOI:** 10.3390/md22040173

**Published:** 2024-04-12

**Authors:** Andrea Defant, Giacomo Carloni, Nicole Innocenti, Tomaž Trobec, Robert Frangež, Kristina Sepčić, Ines Mancini

**Affiliations:** 1Laboratory of Bioorganic Chemistry, Department of Physics, University of Trento, Via Sommarive 14, 38123 Trento, Italy; giacomo.carloni@pasteur.fr (G.C.); nicole.innocenti@unitn.it (N.I.); 2Unit of Structural Microbiology, Pasteur Institute, CNRS, University of Paris City, 75015 Paris, France; 3Institute of Preclinical Sciences, Veterinary Faculty, University of Ljubljana, Gerbičeva 60, 1000 Ljubljana, Slovenia; tomaz.trobec@vf.uni-lj.si (T.T.); robert.frangez@vf.uni-lj.si (R.F.); 4Department of Biology, Biotechnical Faculty, University of Ljubljana, Jamnikarjeva 101, 1000 Ljubljana, Slovenia; kristina.sepcic@bf.uni-lj.si

**Keywords:** marine metabolite, drug design, organic synthesis, molecular docking, ADME prediction, acetylcholinesterase inhibition, Alzheimer’s disease

## Abstract

In this study, Antarctic *Latrunculia* sponge-derived discorhabdin G was considered a hit for developing potential lead compounds acting as cholinesterase inhibitors. The hypothesis on the pharmacophore moiety suggested through molecular docking allowed us to simplify the structure of the metabolite. ADME prediction and drug-likeness consideration provided valuable support in selecting 5-methyl-2H-benzo[h]imidazo[1,5,4-de]quinoxalin-7(3H)-one as a candidate molecule. It was synthesized in a four-step sequence starting from 2,3-dichloronaphthalene-1,4-dione and evaluated as an inhibitor of electric eel acetylcholinesterase (eeAChE), human recombinant AChE (hAChE), and horse serum butyrylcholinesterase (BChE), together with other analogs obtained by the same synthesis. The candidate molecule showed a slightly lower inhibitory potential against eeAChE but better inhibitory activity against hAChE than discorhabdin G, with a higher selectivity for AChEs than for BChE. It acted as a reversible competitive inhibitor, as previously observed for the natural alkaloid. The findings from the in vitro assay were relatively consistent with the data available from the AutoDock Vina and Protein-Ligand ANTSystem (PLANTS) calculations.

## 1. Introduction

Discorhabdin alkaloids are a singular class of marine pigments. Since the first isolation of discorhabdin C in 1986 from a New Zealand sponge [1], these metabolites have been exclusively found in demosponges, mostly Latrunculiidae. They belong to the pyrroloiminoquinone class, which includes dozens of members characterized by structural varieties due to the presence of bromine atoms, thioether moieties, and also oligomeric structures [2,3]. Due to the relevant biological activities shown by some discorhabdin alkaloids (cytotoxic, antibacterial, antiviral, antimalarial, immunomodulatory, and caspase-inhibitory activities), their isolation and structural elucidation have attracted considerable attention over the years [2]. Attention has even been focused on their production through a series of synthetic approaches [4].

In our previous report, discorhabdin G, 3-dihydro-7,8-dehydro discorhabdin C, discorhabdin B, and discorhabdin L (Figure 1), isolated from two specimens collected in the Antarctic region, were characterized for the first time as cholinesterase inhibitors [5]. We considered both acetylcholinesterase (AChE) and butyrylcholinesterase (BChE) due to their physiological roles as therapeutic targets involved in the symptoms of Alzheimer’s disease, in the treatment of which the currently available drugs are AChE inhibitors. The discorhabdin alkaloids tested in our previous work showed inhibition constant values between 1.6–15.0 µM and 5.0–76.0 µM for electric eel AChE (eeAChE) and horse serum BChE, respectively. Among the tested metabolites, discorhabdin G (**1**) was the most active, with IC_50_ values lower than those of physostigmine. We also found that all tested discorhabdin alkaloids act as reversible, competitive inhibitors of selected cholinesterases. Notably, an additional electrophysiological study on discorhabdin G showed no adverse effects on neuromuscular transmission or skeletal muscle function [5]. This finding is particularly promising as this property avoids the side effects that can occur in patients after treatment with some currently used drugs acting as AChE inhibitors. Additionally, molecular docking calculations allowed us to identify the interactions of discorhabdin G with the enzyme’s active site, leading to results consistent with the experimental data of AChE inhibition. Moreover, we preliminarily identified the structural unit of discorhabdin G involved in the interaction with the enzyme, allowing us to hypothesize the pharmacophore moiety [5].

The combination of good AChE inhibitory activity, the absence of possible side effects on the peripheral neuromuscular system, and the preliminary identification of the putative pharmacophore moiety make the most active discorhabdin G a promising lead for further investigations. Our research in this direction was also encouraged by a recent report on the use of marine compounds to treat neurodegenerative diseases, which has been defined as both an opportunity and a challenge [6].

The present work aims to identify the minimal discorhabdin pharmacophore moiety that preserves bioactivity and its use through a synthetic approach that can reduce the structural complexity of the natural molecule. We report here on (i) computationally assisted pharmacophore modeling through docking calculation, (ii) design and chemical synthesis, and (iii) experimental AChE inhibition of simplified molecules related to the structure of discorhabdin G. This work also includes a comparison between the candidate molecules and natural alkaloids based on Absorption, Distribution, Metabolism, and Excretion (ADME)/toxicity prediction and drug-likeness consideration.

## 2. Results and Discussion

### 2.1. Pharmacophore Modeling of Discorhabdin G and Design of Simplified Structures

Our approach to computer-aided drug design started by identifying the pharmacophore unit of discorhabdin G available by molecular docking of its complex with AChE. We provided a first indication from the calculation performed for the natural product in the complex with *Torpedo californica* AChE (1DX6) [2]. At present, an updated version of AutoDockVina is accessible, and recent data obtained by X-ray diffraction analysis with a better resolution are available for *Torpedo californica* AChE (6G1V). We also conducted an in-depth study using the Protein-Ligand ANTSystem (PLANTS) program (detailed below). Through this computational analysis, we can select the best interactions, as visualized in Figure 2. From here, it can be inferred that the brominated ring A and the spiro-bicyclic unit containing the A and B rings (Figure 1) are not involved in the interactions inside the enzyme active site.

In a synthetic approach, structural simplification is permissible, which benefits from overlooking the discorhabdin G moiety that is not involved in the enzyme interaction. Furthermore, this reduced scaffold minimizes the possibility of cytotoxicity, which is undesirable in anti-AChE potential applications. The reported investigation of structure–activity relationships on several discorhabdin members and analogs led to the knowledge of the crucial structural features affecting cytotoxicity. In detail, cytotoxicity was reported to correlate with the electrophilic reactivity of the spirodienone moiety containing the C-3 carbonyl group, with the presence of bromine atom(s), as well as with the thioether bridge (e.g., in discorhabdins B and L). In particular, discorhabdin L (**4**, Figure 1), which showed the lowest eeAChE inhibition among the four discorhabdin members tested in our previous study, is reported to significantly inhibit prostate tumor growth [7].

Furthermore, it is worth pointing out the detrimental effect on the antitumor activity of a double bond in the C7/C8 position [8]. Notably, in recent years, dimers [9,10] and trimers [10] have also been reported to have often displayed good cytotoxicity. However, it is also known that, despite the potent in vitro activity obtained for selected discorhabdin alkaloids (e.g., discorhabdin A) against some human tumor cell lines, they were ineffective in model mouse studies [11].

Discorhabdin G was first isolated in 1995 from an Antarctic collection of *Latrunculia apicalis*, reported as antibacterial and active in causing feeding deterrence behavior in one of the major Antarctic sponge predators [12]. The cytotoxicity of this compound is considerably lower than that of other discorhabdins, as reported for a large library of discorhabdin alkaloids (including discorhabdin G) assessed for effects on Merkel cell carcinoma viability, which revealed no apparent mechanistic differences between the different discorhabdin metabolites tested. The results of this study suggest that these compounds do not induce apoptosis but rather mitochondrial dysfunction, leading to non-apoptotic cell death [11].

Based on this evidence, we designed molecules related to discorhabdin G (**1**) to simplify their structural complexity in favor of more favorable synthetic accessibility. This approach was informed by our previous work on synthesizing heteroaromatic fused compounds from dichlorquinoline-5,8-dione and dichloro-1,4-naphthoquinone [13,14,15]. We evaluated structures **2** and **3** as candidate molecules (Figure 3). Structure **3** presents the scaffold of the natural hit interested in the AChE interaction as deduced by calculations. Nonetheless, we opted to incorporate molecule **2** into the virtual screening due to nitrogen’s role as a proton acceptor, which can influence the cholinesterase active site, a phenomenon observed in several natural alkaloids exhibiting AChE inhibition [16]. In making this choice, we referred to the pharmacophore-directed retrosynthesis (PDR) approach inspired by natural product scaffolds, recently reported by Truax and Romo [17]. This strategy involves developing a pharmacophore hypothesis with more approximate consideration than one encompassing all ligand–receptor interactions.

Initially, we examined compounds **2** and **3** in both their neutral and protonated states, considering pH variations using the MarvinSketch software (version number: 23.17.0) [18]. The analysis indicated the forms likely present at pH 7.4, which mimics the physiological environment for in vitro screening. Interestingly, the predominant forms for 2 and 3 were found to be neutral (99.964% and 99.955%, respectively), while for discorhabdin G, 83.2% of the species were predicted to be protonated and only 16.8% neutral (Appendix A).

Regarding the interaction with *Torpedo californica* AChE (6G1V), docking calculations revealed a hydrogen bond between TYR 334 and the carbonyl group of **3** (Figure 4), besides the TRP 84 and PHE 330 interactions also involved in the complex with discorhabdin G (Figure 2).

### 2.2. ADME/Toxicity Prediction

To support the choice of structures **2** and **3** as candidate molecules, we evaluated their physicochemical parameters and drug-likeness in comparison with discorhabdin G. ADME (Absorption, Distribution, Metabolism, Excretion) prediction was obtained by using Swiss-ADME [19,20] and Molsoft L.C.C. [21] software, applied to the species present at the physiological pH value.

All compounds displayed favorable physicochemical properties, as visualized in the bioavailability radar (Figure 5).

In particular, compound **3** shows the best topological polar surface area (TPSA) value. It is a useful descriptor to estimate the capability to cross the blood–brain barrier (BBB), primarily for a valuable drug in treating Alzheimer’s disorder [19]. The brain or intestinal estimated permeation method (BOILED)-Egg diagram visualizes the lipophilicity (WLOGP) and TPSA correlation for the three compounds and highlights the best BBB permeation capability of compound **3** (Figure 5). The highest value predicted by Molsoft L.L.C. for this molecule supports this finding (Table 1). Aside from this descriptor, gastrointestinal absorption is another pharmacokinetic parameter that results in a positive. Compliance with Lipinski’s rule by all compounds is a favorable evaluation of drug-likeness. Additionally, Molsoft L.L.C. predicted similar values for the drug-likeness model score for **2** and **3** (0.38 and 0.37, respectively), both better than the value for the natural metabolite (0.12) (Appendix A).

### 2.3. Synthesis of Compound ***3***

Based on comparable docking results obtained for **2** (as detailed below) and a more favorable response for **3** in ADME prediction and drug-likeness, we focused on this latter molecule based on a more accessible synthetic approach. This choice is common in medicinal chemistry, where a compromise is often considered among several properly balanced parameters [22].

Figure 1 reports the synthetic sequence planned for obtaining **3**. The same procedure could be applied to the production of **2**, starting from 6,7-dichloroquinoline-5,8-dione [13,14] to yield a mixture of regioisomeric products by treating with ammonium hydroxide, followed by a not-easy chromatographic separation of the desired 7-amino-6-chloroquinoline-5,8-dione.

Compound **3** was obtained by a four-step synthesis, starting from 2,3-dichloronaphthalene-1,4-dione (**4**). After treatment with an aqueous ammonium hydroxide solution in ethanol under heating and stirring, **4** was converted into 2-amino-3-chloronaphthalene-1,4-dione (**5**). We optimized the subsequent production of **6** in high yield by reacting **5** and an equimolar amount of acetic anhydride in the presence of catalytic sulfuric acid under sonication for 15 min. Conversely, using a significant excess of acetic anhydride in the presence of a catalytic amount of concentrated sulfuric acid provided acetamido derivative **6** as a minor compound and 2-methylnaphtho[2,3-d]oxazole-4,9-dione (**7**) as the main product by the cyclization reported for amide **6** [23]. The tetrahydropirazine ring of product **8** was introduced by stirring compound **6** in acetonitrile with an equimolar amount of ethylendiamine at room temperature. This cyclization became more complex than expected, posing a challenge to obtaining the target compound in high yield. Still, at the same time, it allowed elucidation of the reaction mechanism. The isolation of the intermediate product enabled us to see that the cyclization occurs through the amine substitution of the chloride first, followed by the attack of the primary amino group of the (2-aminoethyl)amino moiety on the carbonyl group, releasing water. As the last step of the sequence, heating compound **8** in ethanol with acetyl chloride gave fully conjugated **9** as the exclusive product. The treatment of **8** in solution with glacial acetic acid and the simultaneous hydrogen flow in the presence of Pd/C yielded target product **3**, which was obtained as an acetate salt.

The products of each step were chromatographically purified when necessary and structurally characterized by ESI-MS and ^1^H and ^13^CNMR analyses, further supported by long-range hetero-correlations from HMBC experiments. An HPLC analysis allowed us to establish that the purity of compounds **3**, **7**, and **8**, later subjected to in vitro screening, was higher than 95% (Appendix A).

### 2.4. Biological Evaluation

Table 2 reports the data on enzyme inhibition obtained for the synthetic compounds, including the designed compound **3**, its synthetic precursor **8**, and the fused tricyclic compound **7** (Figure 1). In particular, we selected **7** for its naphthoquinone structure based on the knowledge that several quinones efficiently inhibit not only AChE but also amyloid-β (Aβ) aggregation, which is an attractive feature for the development of multitarget drugs for the symptomatic treatment of Alzheimer’s disease [24]. These also include natural products, an example of which is the quinone derivative anhydrojavanicin, isolated from an extract of the fungus *Aspergillus terreus* [25].

Physostigmine salicylate, which was used as a positive control, is a drug that was used in patients with Alzheimer’s disease and has recently been replaced by other AChE inhibitors due to its poor bioavailability and side effects. The degree of cholinesterase inhibition by physostigmine salicylate observed in this study somehow differs from that detected during our previous evaluation of natural discorhabdin alkaloids under the same experimental conditions [2]. However, this is not surprising in light of what is reported in the literature, where it is not unusual to encounter 10-100-fold differences in binding affinities of the same compound for the same type of cholinesterase, making accurate and direct comparisons between studies complex [25]. In our case, this difference could be attributed to different enzyme lots used over time. Therefore, we selected neostigmine methylsulfate as an additional reference. The documented comparable AChE inhibition by physostigmine salicylate and neostigmine methylsulfate [25,26], also observed in our assay (Table 2), can support our explanation.

Typically, patients with Alzheimer’s disease first show enhanced activity of AChE in some regions of the brain and consequently suffer from an acetylcholine deficit and loss of cognitive functions. As Alzheimer’s disease progresses, the AChE levels in the brain start to decrease and are compensated by an increased activity of BChE. Therefore, developing dual (AChE and BChE) inhibitors is encouraged [27]. Further, only reversible cholinesterase inhibitors can have a practical therapeutic application because irreversible inhibitors can be lethal. In particular, galantamine and donepezil are drugs involved in transient bonding with both cholinesterases and thus act through a reversible mechanism. In contrast, rivastigmine inhibits AChE in a pseudo-irreversible, non-competitive manner [16].

In this study, we tested the inhibition potential of the compounds against eeAChE, human recombinant AChE (hAChE), and horse serum BChE. Product **3** exhibited the highest inhibitory potential (Appendix A) and the lowest IC_50_ values in the series against the two AChE enzymes, with inhibition decreasing drastically by moving to **7** and **8**. A kinetic analysis showed that both **3** and **7** exhibited a reversible competitive type of inhibition against all tested cholinesterases (Appendix A). The same inhibition type was determined for discorhabdin G [5], indicating the interaction of these compounds with the enzymes’ active sites. Compared with discorhabdin G, **3** showed a slightly lower inhibitory potential against eeAChE but better inhibitory activity against hAChE and a higher selectivity for AChEs related to BChE.

### 2.5. Molecular Docking Study

*Torpedo californica* AChE was adopted as a model system in virtual screening based on the high similarity of aminoacidic residues in the active sites to the eeAChE used in the biological assays, as already obtained using the TM align algorithm [5]. In addition to the previously considered *Torpedo californica* AChE complexes with galantamine (1DX6, 2.3 Å resolution) [5], we performed the calculations on *Torpedo californica* AChE (6G1V, 1.82 Å) complexed with 12-amino-3-chloro-6,7,10,11-tetrahydro-5,9-dimethyl-7,11-methanocycloocta[b]quinolin-5-ium, now available with a better resolution (1.82 Å). Compounds **3**, **2**, and **7** were considered ligands in comparison with discorhabdin G (**1**) in its protonated form, as supported by the significant form predicted at physiological pH and based on previous data [5]. The results are given as energy values in kcal/mol for AutoDock Vina and as PLANTS scores (Table 3). All data obtained for discorhabdin G (**1**) show a more favorable virtual inhibition than those obtained for **2**, **3**, and **7**. Both PLANTS and Vina calculations provided the same trend in favor of compound **3** compared to analog **2**. Besides the finding of more favorable results for **3** than **2**, the interactions visualized in Figure 3 support the choice of molecule **3** for synthesis and experimental evaluation. The complex of **7** with *T. californica* AChE (6G1V) showed the same hydrogen bond involving TYR334 and π–π interactions with PHE 330 and TRP 84 (Appendix A) as were observed for **3** (Figure 4); however, it was associated with a less favorable energy value by both docking programs. The highest energy values and scores provided by **7** in the complexes with the two *T. californica* AChEs (Table 3) are in line with its experimental data. Fewer interactions, especially any missing hydrogen bond as indicated by docking calculations (Appendix A), support the behavior of compound **8**, resulting in it being the most inactive in the in vitro assay. Figure 6 shows the overlapping of ligand **3** as the most active synthetic compound and the natural product **1**.

The Vina energy values obtained for each complex of discorhabdin G, **3**, **7**, and physostigmine with *Homo sapiens* AChE (4M0E) are all within a deviation of 2 kcal/mol, which makes the results not significantly different (Table 3). Therefore, we found no decisive correlation between the computational data and the results of the enzyme inhibition assay.

## 3. Materials and Methods

### 3.1. Chemistry

#### 3.1.1. General Experimental Procedures

All chemicals and reagents were purchased from Sigma Aldrich (Taufkirchen, Germany). The yields were calculated on the purified products unless otherwise indicated. Thin-layer chromatography (TLC) was performed using Merck silica gel F_254_, using short-wave UV light as the visualizing agent, and cerium sulfate was used as a developing agent upon heating. Preparative thin-layer chromatography (PLC) was performed using 20 × 20 cm Merck Kieselgel 60 F_254_ 0.5/2-mm plates. Column chromatography was performed using Merck Si 45–60 µm as the stationary phase. The purity of compounds **3**, **7**, and **8** used for bioassays was determined to be >95% by High-Performance Liquid Chromatography (HPLC) using an Agilent 1200 HPLC system (Agilent Technologies, Waldbroon, Germany)equipped with an autosampler, a binary pump, a diode array detector (Agilent Technologies), and a reversed-phase column (Phenomenex Gemini 5 µm C18 110A), under gradient conditions with eluent water/acetonitrile (CH_3_CN t_0_ 30%, t_8 min_ 80%, and t_22 min_ 80%) at a flow rate of 1 mL/min and a λ of 254 nm. NMR spectra were recorded on a Bruker Avance 400 spectrometer using a 5 mm BBI probe ^1^H at 400 MHz and ^13^C at 100 MHz and calibrated using residual undeuterated solvent for CDCl_3_ (relative to δ_H_ 7.25 ppm and δ_C_ 77.0 ppm, respectively) with chemical shift values in ppm and *J* values in Hz. NMR data were analyzed using Bruker TopSpin software, version 3.6.1. Assignments were made by heteronuclear multiple bond correlation (HMBC) experiments. ^1^H and ^13^C NMR spectra of the tested compounds are reported in Appendix A. MS and tandem (MS/MS)^n^ were taken through a Bruker Esquire-LC mass spectrometer equipped with an electrospray ionization (ESI) source in positive and negative ion modes. The sample was injected into the source from a methanolic solution. High-resolution ESI-MS measurements were obtained by direct infusion of a methanol solution using an Orbitrap Fusion Tribrid mass spectrometer.

#### 3.1.2. Synthesis of 5-Methyl-2H-benzo[h]imidazo[1,5,4-de]quinoxalin-7(3H)-one (**3**)

1.2-Amino-3-chloronaphthalene-1,4-dione (**5**)

To a solution of 2,3-dichloronaphthalene-1,4-dione (**4**, 400 mg, 1.76 mmol) in EtOH heated at 50 °C, 30% (*w*/*w*) ammonia solution (10.0 mL) was added. The mixture was stirred at 50 °C for 40 min, monitored by TLC (hexane/AcOEt 7:3). The solvent was removed under reduced pressure, and the residue was partitioned between the saturated solution of NaHCO_3_ and dichloromethane. The organic phase was dried with anhydrous Na_2_SO_4_ and evaporated to dryness, yielding the pure compound (360 mg, yield 99%).

^1^H-NMR (400 MHz, CDCl_3_) δ: 8.15 (d, *J* = 7.6 Hz, 1H), 8.06 (d, *J* = 7.6 Hz, 1H), 7.74 (td, *J* = 7.6, 1.3 Hz, 1H), 7.65 (td, *J* = 7.7, 1.5 Hz, 1H). ESI(+)-MS: *m*/*z* 207.9 [M + H]^+^; ESI(-)-MS: *m*/*z* 206 [M−H]^−^.

2.*N*-(3-Chloro-1,4-dioxo-1,4-dihydronaphthalen-2-yl)acetamide (**6**) and byproduct **7**

A solution of **5** (20 mg, 0.096 mmol) in acetic anhydride (0.5 mL, 5.3 mmol) was treated with a catalytic amount of 98% H_2_SO_4_ and monitored by TLC (hexane/AcOEt 7:3). After 30 min, water was slowly added to hydrolyze acetic anhydride in excess. The mixture was neutralized with a saturated solution of NaHCO_3_ and extracted with ethyl acetate (x3). The combined organic phases were concentrated in vacuo and purified by PLC (hexane/AcOEt 7:3), obtaining two bands corresponding to tricyclic compound **7** in a significant amount (15 mg, yield 73%) and compound **6** (2 mg, yield 10%). Therefore, we optimized the previous procedure to obtain **6** as the primary product. An equimolar mixture of compound **5** (100 mg, 0.48 mmol), acetic anhydride (45 µL, 0.48 mmol), and a catalytic amount of concentrated H_2_SO_4_ was sonicated for 15 min. The mixture was partitioned between water and ethyl acetate. The organic phase was dried on anhydrous Na_2_SO_4_ and concentrated in vacuo. The residue was subjected to chromatographic purification by gradient elution with hexane/AcOEt, obtaining pure compound **6** as a yellow powder (90 mg, yield 88%).

Data for **6**: ^1^H-NMR (400 MHz, CDCl_3_) δ: 8.17 (dd, *J*= 8.7, 1.8 Hz, 1H), 8.10 (dd, *J*= 8.7, 1.8 Hz, 1H), 7.78 (m, 2H), 7.69 (brs, 1H), 2.30 (s, 3H). ^13^C-NMR (detectable signals) δ: 179.9, 178.0, 170.9, 134.4, 131.4, 130.3, 127.4, 115.0, 23.5. Significant ^1^H,^13^C long-range correlations: δ 8.17 with 178.0 and 130.3 ppm, δ 8.10 with 179.8 and 131.4 ppm, δ 2.30 with 170.9 ppm. ESI(+)-MS: *m*/*z* 272 [M+Na]^+^; ESI(-)-MS: *m*/*z* 248 [M−H]^−^; MS/MS (248): *m*/*z* 233, 212.

Data for 2-methylnaphtho[2,3-d]oxazole-4,9-dione (**7**): HPLC t_R_ = 5.4 min, purity 97.4% (Appendix A). ^1^H-NMR (400 MHz, CDCl_3_) δ: 8.12 (d, *J* = 7.6 Hz, 1H), 7.67 (m, 2H), 7.52 (m, 1H), 2.62 (s, 3H). ^13^C-NMR (100 MHz, CDCl_3_) δ: 179.8, 172.2, 163.6, 159.0, 135.5, 134.3, 131.11 (2C), 129.3, 126.0, 122.8, 14.2. Significant ^1^H,^13^C long-range correlations: δ 8.12 and 7.67 with 179.8, 159.0, and 126.0 ppm; δ 2.62 with 163.6 ppm. ESI(+)-MS: *m*/*z* 214 [M + H]^+^, 236 [M + Na]^+^.

3.N-(6-oxo-2,3,4,6-tetrahydrobenzo[f]quinoxalin-5-yl)acetamide (**8**)

To a solution of **6** (60 mg, 0.24 mmol) in CH_3_CN (2 mL), ethylendiamine (16 µL, 0.24 mmol) was added. The mixture was stirred at room temperature for 4 h, monitoring the reaction by TLC (CH_2_Cl_2_/MeOH 95:5). The solvent was evaporated under reduced pressure, and the residue was purified by FC, eluting with CH_2_Cl_2_/MeOH from 99:1 to 94:6 to obtain a red product (38.5 mg, yield 63%).

HPLC: t_R_ = 4.2 min, purity 95.8% (Appendix A). ^1^H-NMR (400 MHz, CDCl_3_) δ: 8.16 (m, 2H), 8.05 (dt, *J* = 8.0, 3.1 Hz, 1H), 7.54 (m, 2H), 7.46 (brs, 1H), 4.12 (t, *J* = 6.4 Hz, 2H), 3.39 (td, *J* = 6.5, 2.8 Hz), 2.25 (s, 3H). ^13^C-NMR (100 MHz, CDCl_3_) δ: 178.2, 169.3, 153.7, 134.3, 133.2, 132.1, 130.8, 130.2, 125.6, 123.9, 112.2, 47.9, 37.0, 24.2. ESI(+)-MS: *m*/*z* 256 [M + H]^+^, 278 [M + Na]^+^; MS/MS (256): *m*/*z* 214; MS^3^ (214): *m*/*z* 197; ESI(-)-MS: *m*/*z* 254 [M−H]^−^; MS/MS (254): *m*/*z* 211.

4.Final product **3** and compound **9**

To a solution of **8** (10.0 mg, 0.039 mmol) in absolute ethanol (2 mL), acetyl chloride (10 µL, 0.15 mmol) was added, and the mixture was refluxed for 1 h. After evaporation of volatiles, the residue was subjected to a preparative TLC (CH_2_Cl_2_/MeOH 95:5), which yielded pure **9** (5.0 mg, yield 55%). To obtain the desired compound **3**, hydrogen produced by a suitable apparatus was flowed into a boiling solution of **8** (15 mg, 0.059 mmol) in glacial acetic acid (0.75 mL) in the presence of 10% palladium on carbon for one hour, monitoring the product formation by TLC (CH_2_Cl_2_/MeOH/Et_3_N 90:9:1). The solution was filtered to remove the palladium catalyst, and acetic acid was evaporated under reduced pressure to yield pure **3** as a light-green solid, isolated as acetate salt (7.2 mg, yield 41%). A portion was treated with Et_3_N and eluted through a column of silica gel using hexane/ethyl acetate 7:3 to obtain **3** in neutral form.

Data for 5-methyl-7H-benzo[h]imidazo[1,5,4-de]quinoxalin-7-one (**9**): ^1^H-NMR (400 MHz, CDCl_3_) δ: 8.64 (m, 1H), 8.56 (m, 1H), 8.15 (m, 1H), 7.89 (m, 1H), 7.78 (m, 2H), 2.87 (s, 3H). ESI(+)-MS: *m*/*z* 235 [M + Na]^+^.

Data for 5-methyl-2H-benzo[h]imidazo[1,5,4-de]quinoxalin-7(3H)-one (**3**): HPLC (on compound after treating with Et_3_N): t_R_ = 4.7 min, purity 95.7% (Appendix A). ^1^H NMR (400 MHz, CDCl_3_) δ: 8.31 (dd, *J* = 6.0, 3.2 Hz, 1H), 8.21 (dd, *J* = 5.5, 3.5 Hz, 1H), 7.63 (dd, *J* = 5.8, 3.4 Hz, 2H), 4.41 (t, *J* = 7 Hz, 2H), 4.12 (t, *J* = 7 Hz, 2H), 2.53 (s, 3H), 2.09 (s, 3H, CH_3_COOH). Significant ^1^H,^13^C long-range correlations: δ 2.09 (CH_3_COOH) with 177.8 ppm. ^13^C-NMR on neutral **3** (100 MHz, CDCl_3_) δ: 175.4, 165.1, 152.0, 132.5, 131.7, 128.8, 127.0, 124.6, 109.9, 48.3, 39.4, 13.9. HRESI(+)-MS: *m*/*z* 238.09788 ± 0.0005, calcd. for C_14_H_12_N_3_O: 238.09749. ESI(+)-MS: *m*/*z* 238 [M + H]^+^, 260 [M + Na]^+^; MS/MS (238): *m*/*z* 223, 211, 197; ESI(-)-MS (after treating with Et_3_N) *m*/*z*: 236 [M−H]^−^, MS/MS (236): *m*/*z* 221, 209.

### 3.2. Biological Evaluation

The inhibition of cholinesterase activity was determined using a modification of the Ellman method [28], which was adapted for microtiter plates. Stock solutions of **3**, **7**, and **8** (2 mg/mL) were prepared in pure EtOH. Positive controls (neostigmine methylsulfate (2 mg/mL) and physostigmine salicylate (5 mg/mL), both from Sigma-Aldrich, St. Louis, MO, USA) were also prepared in EtOH. Stock solutions of the potential inhibitors and the positive controls were added to the microtiter plate wells and progressively diluted in 100 mM potassium phosphate buffer (pH 7.4) to a final volume of 50 μL. Then, 100 μL acetylthiocholine chloride (1 mM) and 5,5′-dithiobis-2-nitrobenzoic acid (0.5 mM) in 100 mM potassium phosphate buffer (pH 7.4) was added into the microtiter plate wells. The cholinesterases that were used (eeAChE, hAChE, and BChE; all from Sigma-Aldrich, St. Louis, MO, USA) were dissolved in 100 mM potassium phosphate buffer (pH 7.4) to a final concentration of 0.0075 U/mL. Each cholinesterase solution (50 μL) was added into the microtiter plate wells to start the reaction, which was followed spectrophotometrically at 405 nm at 25 °C over 5 min using a kinetic microplate reader (Dynex Technologies Inc., Chantilly, VA, USA). Blank reactions without inhibitors were run with the appropriate dilutions of EtOH, and the readings were corrected according to the corresponding blanks. Each measurement was repeated at least three times. Inhibitory constants (*K_i_*) were determined by following the kinetics using three different final substrate concentrations (0.125, 0.25, 0.5 mM). The data were analyzed using the OriginPro software (OriginPro 2021 (9.8), OriginLab Corporation, Northampton, MA, USA).

### 3.3. Computational Details

#### 3.3.1. Prediction of Structures at Different pH Values

MarvinSketch 23.17.0 was used for the prediction of the molecular structures of **2** and **3** at different pH values, developed by ChemAxon [18].

#### 3.3.2. Pharmacokinetic Study

ADME predictions were performed using the Swiss-ADME online server [19,20] and Molsoft L.L.C. [21].

#### 3.3.3. Docking Calculation

Calculations were carried out on a PC running at 3.4 GHz on an AMD Ryzen 9 5950X on an home assembled PC with 16-core (32 threads) processor with 32 GB RAM and 1 TB hard disk with Windows 10 Home 64-bit as an operating system. The ligands were built using PC Model version 10 (Serena Software 10.074000, Bloomington, IN 47402e3076) and pre-minimized using the force field MMX. The pre-minimized molecules were subjected to geometry optimization at the density functional theory (DFT) level in the gas phase. The optimized geometry was obtained by using the RFO step, integral precision=superfine grid, and type convergence criteria, invoking the gradient employing the 6-31+G(d,p) basis set for C, H, N, and O atoms and the extra basis set 6-311+G(3df) for bromine atoms. The electronic correlation functional B1B95, which combines the gradient-corrected DFT with Becke hybrid functional B1 [29] for the exchange part and B95 for the correlation function [30], was utilized. The vibrational energy calculations at the DFT levels used the optimized structural parameters to characterize all stationary points as minima.

No imaginary wave-number modes were obtained for the optimized structure, proving that a local minimum on the potential energy surface was actually found. Each ligand output file, deriving from DFT calculations, was converted by open Babel version 3.1.1 [31] to a pdbqt file, taking quantum mechanical Mulliken charges instead of using the Gasteiger–Marsili charges for further docking calculations. The AutoDock Tools (ADT) package version 1.5.6rc3 [32] was used to generate the docking input files for the docking calculations. The different crystallographic structures of AChE were from the Protein Data Bank (PDB; http://www.pdb.org/, accessed on 3 January 2024); the structures of *Torpedo californica* AChE complexed with 12-amino-3-chloro-6,7,10,11-tetrahydro-5,9-dimethyl-7,11-methanocycloocta[b]quinolin-5-ium (6G1V) and with (-)–galantamine (1DX6), with a resolution of 1.82 Å and 2.30 Å, respectively [33,34], and the structure of *Homo sapiens* AChE in complex with dihydrotanshinone (4M0E), with a resolution of 2.0 Å, were all determined by X-ray crystallography [35]. The structures were modified as follows: the ligand and all crystallization water molecules were removed, with the file saved in pdb extension. All hydrogen atoms were added using AutoDock Tools (ADT), and the Gasteiger–Marsili charges were calculated, with the resulting file saved in pdbqt extension. Rotatable bonds were defined for each minimized ligand molecule. Two different approaches were used for the docking calculations: Vina 1.2.3 [36], which employs a genetic algorithm, and PLANTS, which uses a class of stochastic optimization algorithms called ant colony optimization (ACO) [37]. For the Vina calculations, a grid box of 14 × 14 × 14 Å in the x, y, and z directions was created with spacing of 1.00 Å and centered at x = 3.697, y = −4.587, z = 20.236 for 6G1V and at x = 12.023, y = −52.628, z = −23.394 for 4M0E. Further parameters were set as follows: exhaustiveness of the local search, 100; number of conformations to calculate, 10. To validate the goodness of the calculation, the original ligand was re-docked, and visual inspection of the data showed a very tight overlap. The results are expressed as the energy associated with each ligand–enzyme complex in terms of the Gibbs free energy values. For the PLANTS calculations, the structure of the enzyme and the ligands were saved in mol2 extension; a sphere with radius = 12 Å was centered at the same position used for Vina, and the chemPLP scoring function was employed [38], saving 10 cluster structures with RMSD = 2.00 Å. Each Vina and PLANTS docked complex was re-minimized using MMFF94x force field. The visual inspection of the ligand–enzyme interactions was displayed using Biovia Discovery Studio visualizer (Discovery Studio Visualizer v21.1.0.20298) [39].

## 4. Conclusions

This work follows the idea that simplified structures of bioactive natural products have the potential to serve as starting points for the development of new drugs suitable for symptomatic treatment of the early stages of Alzheimer’s disease. We evaluated marine sponge-derived alkaloid discorhabdin G as a hit for developing potential lead compounds acting as cholinesterase inhibitors. This was achieved starting from the hypothesis on the pharmacophore of the metabolite, suggested through molecular docking of the AChE–metabolite complex. The evaluation of ADME prediction provided valid support in selecting candidate molecule **3**, which we synthesized in a four-step sequence starting from 2,3-dichloronaphthalene-1,4-dione. Its four-ring structure was essential for bioactivity, as established by an in vitro assay compared to related molecules accessible by its synthetic sequence and the known data for the natural metabolite. Compounds **3** and **7** exhibited a reversible competitive cholinesterase inhibition, as previously observed for discorhabdin G. A lower inhibition of eeAChE but a higher activity on hAChE and higher selectivity for AChEs than for BChE were observed for compound **3** as compared to discorhabdin G. The molecular docking calculations performed by the AutoDock Vina and PLANTS tools on discorhabdin G and compounds **3** and **7** in complexes with both *Torpedo californica* and human AChE helped support most of the experimental data on bioactivity. In summary, this work has efficiently provided potential starting data to be improved towards identifying increasingly active simplified structures inspired by natural discorhabdin G. Importantly, the results obtained for compound **3** are relevant to support the efficacy of the approach adopted and serve as a starting point for lead optimization after assessing its possible cytotoxicity as well as its putative undesirable effects on neuromuscular transmission and skeletal muscle.

## Data Availability

The original data presented in the study are included in the article/Appendix A; further inquiries can be directed to the corresponding author.

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
