# Peer review of "Structural Insights into the Marine Alkaloid Discorhabdin G as a Scaffold towards New Acetylcholinesterase Inhibitors"

_marinedrugs, 2024, doi:10.3390/md22040173_

Round 1

Reviewer 1 Report

Comments and Suggestions for Authors

In their manuscript "Structural Insights into the Marine Alkaloid Discorhabdin G as Scaffold Towards New Acetylcholinesterase Inhibitors," Defant et al. explore the design space of natural product-based discorhabdin analogues in the context of acetylcholine esterase (AChE) inhibitor search. Inhibition of AChE proved useful for the treatment of neurogenerative conditions and for many other applied biochemistry fields as well. The use of natural products as drugs and drug leads experiences an exponential rise nowadays. Thus, the study is based on strong scientific grounds and would be interesting to a wide range of researchers and healthcare professionals.
The authors found no clear correlation between docking scores and the in vitro activity of the compounds studied. Although negative in some sense, these results are interesting to the drug design community. The higher selectivity of compound 3 toward hAChE over BuChE comprises a significant finding of this study.

I have several minor comments to the article:
- figure 5 quality is poor (might be due to pdf image compression settings). It may be useful to re-make these figures in the R plotly package or sigmaplot;
- figure S2, S6, and S7 are also far from perfect, which might not be a great deal since they are placed in supplementary material;
- line 223 active site titration in the lab could improve the situation;
- line 406 "appro-priate".

Since I have found only minor and rather subjective inconsistencies in the manuscript, the paper can, in my opinion, be published as it is, but I would recommend addressing these technical points to polish the paper at the final stages.

Author Response

We are very grateful to the reviewer for the time devoted to this revision, as well as for the valuable feedback and constructive comments on the manuscript.

The changes are currently highlighted within the new version of the manuscript. Please find below, point-by-point our responses to the specific comments and suggestions.

In their manuscript "Structural Insights into the Marine Alkaloid Discorhabdin G as Scaffold Towards New Acetylcholinesterase Inhibitors," Defant et al. explore the design space of natural product-based discorhabdin analogues in the context of acetylcholine esterase (AChE) inhibitor search. Inhibition of AChE proved useful for the treatment of neurogenerative conditions and for many other applied biochemistry fields as well. The use of natural products as drugs and drug leads experiences an exponential rise nowadays. Thus, the study is based on strong scientific grounds and would be interesting to a wide range of researchers and healthcare professionals.
The authors found no clear correlation between docking scores and the in vitro activity of the compounds studied. Although negative in some sense, these results are interesting to the drug design community. The higher selectivity of compound 3 toward hAChE over BuChE comprises a significant finding of this study.

I have several minor comments to the article:

(1) figure 5 quality is poor (might be due to pdf image compression settings). It may be useful to re-make these figures in the R plotly package or sigmaplot;

From the authors: We replaced the figure with a new version

(2) figure S2, S6, and S7 are also far from perfect, which might not be a great deal since they are placed in supplementary material;

From the authors: We improved some figures.

(3) line 223 active site titration in the lab could improve the situation;

From the authors: The reviewer is right when saying that enzyme active site titration would lead to more conclusive results. However, the enzyme lots we had used for conducting this study are already consumed and we are unable to reproduce these data. We therefore base our argumentation on documented comparable AChE-inhibitory activity of neostigmine and physostigmine, and we added an additional reference that reinforces this statement:

Soukup O, Winder M, Killi UK, Wsol V, Jun D, Kuca K, Tobin G. Acetylcholinesterase Inhibitors and Drugs Acting on Muscarinic Receptors- Potential Crosstalk of Cholinergic Mechanisms During Pharmacological Treatment. Curr Neuropharmacol. 2017, 15, 637-653. doi: 10.2174/1570159X14666160607212615

(4) line 406 "appro-priate".

From the authors: done. This mistake arised upon conversion of the Word file to the pdf. In the Word file, the word “appropriate” is reported in line 400 (not 406), and there it is separated since its spans two lines of the text.

Since I have found only minor and rather subjective inconsistencies in the manuscript, the paper can, in my opinion, be published as it is, but I would recommend addressing these technical points to polish the paper at the final stages.

Reviewer 2 Report

Comments and Suggestions for Authors

In the study conducted by Andrea et al., the researchers explore the utility of discorhabdin G, sourced from the Antarctic Latrunculia sponge, as a foundational molecule for the development of acetylcholinesterase (AChE) inhibitors, targeting Alzheimer's disease treatments. By employing pharmacophore modeling and molecular docking techniques to streamline discorhabdin G’s structure, the team crafted a candidate molecule, 5-methyl-2H-benzo[h]imidazo[154-de]quinoxalin-7(3H)-one (3). This molecule demonstrated enhanced selectivity in inhibiting human AChE compared to butyrylcholinesterase (BChE). A pivotal observation was that eliminating the spirodienone moiety, which includes the C-3 carbonyl group, augmented this selectivity towards human AChE over BChE.

Some suggestions:

1. The manuscript add more detailed introduction that encompasses the broader context of discorhabdin alkaloids. This should include insights into the initial discovery, the range of biological activities these alkaloids exhibit, and an overview of previous synthetic efforts.

2. The specificity of discorhabdin G for AChE as its primary target merits further investigation. Are there additional potential targets of discorhabdin G that might contribute to its cytotoxic effects?

3. The assertion that removing the spirodienone moiety might diminish cytotoxicity needs substantiation. It would be pertinent to include EC50 data for compound 3 to validate this claim.

4. Comprehensive kinetic studies comparing the interactions of compound 3 with AChE and BChE would enrich the findings, offering a clearer comparison of its efficacy against these enzymes.

Author Response

We are very grateful to the reviewer for the time devoted to this revision, as well as for the valuable feedback and constructive comments on the manuscript.

The changes are currently highlighted within the new version of the manuscript. Please find below, point-by-point our responses to the specific comments and suggestions.

Reviewer 2

In the study conducted by Andrea et al., the researchers explore the utility of discorhabdin G, sourced from the Antarctic Latrunculia sponge, as a foundational molecule for the development of acetylcholinesterase (AChE) inhibitors, targeting Alzheimer's disease treatments. By employing pharmacophore modeling and molecular docking techniques to streamline discorhabdin G’s structure, the team crafted a candidate molecule, 5-methyl-2H-benzo[h]imidazo[154-de]quinoxalin-7(3H)-one (3). This molecule demonstrated enhanced selectivity in inhibiting human AChE compared to butyrylcholinesterase (BChE). A pivotal observation was that eliminating the spirodienone moiety, which includes the C-3 carbonyl group, augmented this selectivity towards human AChE over BChE.

Some suggestions:

1. The manuscript add more detailed introduction that encompasses the broader context of discorhabdin alkaloids. This should include insights into the initial discovery, the range of biological activities these alkaloids exhibit, and an overview of previous synthetic efforts.

From the authors: We inserted additional details with the corresponding references on the different aspects pointed out in the reviewer’s comments.           

In the Introduction: 

Since the first isolation of discorhabdin C in 1986 from a New Zealand sponge [1], these metabolites have been exclusively found in demosponges, mostly Latrunculiidae. They belong to the pyrroloiminoquinone class, including dozens of members characterized by structural varieties due to the presence of bromine atoms, thioether moieties and also oligomeric structures [2, 3]……… …Attention has even been focused on their production by a series of synthetic approaches [4].

In Results and discussion:

In particular, discorhabdin L (4, Figure 1) which showed the lowest eeAChE inhibition among the four discorhabdin members tested in our previous study, is reported to signifi-cantly inhibit prostate tumor growth [7]…………… Notably, dimers [ 9, 10] and trimers [10] have also been reported in recent years to have often displayed good cytotoxicity.

2. The specificity of discorhabdin G for AChE as its primary target merits further investigation. Are there additional potential targets of discorhabdin G that might contribute to its cytotoxic effects?

From the authors: Until now, the cholinesterases seem to be the highest affinity targets for discorhabdin G. The cytotoxicity of this compound is considerably lower comparing with other discorhabdins (Orfanoudaki et al., 2023, refrence already cited), and the electrophysiological experiments showed that it has no undesirable effects on neuromuscular transmission and skeletal muscle at AChE-inhibitory concentrations (Botić et al., 2017).

In the text we added the following details:

Discorhabdin G was first isolated in 1995 from an Antarctic collection of Latrunculia apicalis, reported as antibacterial and active in causing a feeding deterrence behavior on one of the major Antarctic sponge predators [12]. The cytotoxicity of this compound is considerably lower than other discorhabdins, as reported for a large library of discorhab-din alkaloids (including discorhabdin G) assessed for effects on Merkel cell carcinoma viability which revealed no apparent mechanistic differences between different dis-corhabdin metabolites tested. The results of this study suggest that these compounds do not induce apoptosis, but rather induce mitochondrial dysfunction leading to non-apoptotic cell death [11].

3. The assertion that removing the spirodienone moiety might diminish cytotoxicity needs substantiation. It would be pertinent to include EC50 data for compound 3 to validate this claim.

From the authors: We fully agree that the assessment of the cytotoxic potential of the compound 3, that showed promising potential as a drug lead in our study, would be very appropriate. However, our experimental approach was mainly devoted to the assessment of inhibitory potential of the new compounds on three different enzymes and as a consequence, we are lacking the compound 3. However, in the near future we plan to test its cytotoxicity on normal and several transformed cell lines, and compare it with other discorhabdin alkaloids. Also, we plan testing its possible undesirable effects on neurotransmission and skeletal muscle.

 We have emphasized this also in the conclusions:

“Not the least, the results obtained for compound 3 are relevant to support the efficacy of the approach adopted and serve as a starting point for lead optimization after assessing its possible cytotoxicity, as well as its putative undesirable effects on neuromuscular transmission and skeletal muscle.

4. Comprehensive kinetic studies comparing the interactions of compound 3 with AChE and BChE would enrich the findings, offering a clearer comparison of its efficacy against these enzymes.

From the authors: An additional figure (Figure S3) representing the inhibition of the three tested cholinesterases by different concentrations of the compounds 3, 7, and 8 has been included to the Supplementary file and cited in the main text. These graphs clearly demonstrate the highest inhibitory potential of the compound 3 over the compounds 7 and 8, as well as its selectivity for AChEs over BChE.

We are very grateful to the reviewer for the time devoted to this revision, as well as for the valuable feedback and constructive comments on the manuscript.

The changes are currently highlighted within the new version of the manuscript. Please find below, point-by-point our responses to the specific comments and suggestions.

Reviewer 2

In the study conducted by Andrea et al., the researchers explore the utility of discorhabdin G, sourced from the Antarctic Latrunculia sponge, as a foundational molecule for the development of acetylcholinesterase (AChE) inhibitors, targeting Alzheimer's disease treatments. By employing pharmacophore modeling and molecular docking techniques to streamline discorhabdin G’s structure, the team crafted a candidate molecule, 5-methyl-2H-benzo[h]imidazo[154-de]quinoxalin-7(3H)-one (3). This molecule demonstrated enhanced selectivity in inhibiting human AChE compared to butyrylcholinesterase (BChE). A pivotal observation was that eliminating the spirodienone moiety, which includes the C-3 carbonyl group, augmented this selectivity towards human AChE over BChE.

Some suggestions:

1. The manuscript add more detailed introduction that encompasses the broader context of discorhabdin alkaloids. This should include insights into the initial discovery, the range of biological activities these alkaloids exhibit, and an overview of previous synthetic efforts.

From the authors: We inserted additional details with the corresponig refrences on the different aspects pointed out in the reviewer’s comments.           

 In the Introduction: 

Since the first isolation of discorhabdin C in 1986 from a New Zealand sponge [1], these metabolites have been exclusively found in demosponges, mostly Latrunculiidae. They belong to the pyrroloiminoquinone class, including dozens of members characterized by structural varieties due to the presence of bromine atoms, thioether moieties and also oligomeric structures [2, 3]……… …Attention has even been focused on their production by a series of synthetic approaches [4].

In Results and discussion:

In particular, discorhabdin L (4, Figure 1) which showed the lowest eeAChE inhibition among the four discorhabdin members tested in our previous study, is reported to signifi-cantly inhibit prostate tumor growth [7]…………… Notably, dimers [ 9, 10] and trimers [10] have also been reported in recent years to have often displayed good cytotoxicity.

2. The specificity of discorhabdin G for AChE as its primary target merits further investigation. Are there additional potential targets of discorhabdin G that might contribute to its cytotoxic effects?

From the authors: Until now, the cholinesterases seem to be the highest affinity targets for discorhabdin G. The cytotoxicity of this compound is considerably lower comparing with other discorhabdins (Orfanoudaki et al., 2023, refrence already cited), and the electrophysiological experiments showed that it has no undesirable effects on neuromuscular transmission and skeletal muscle at AChE-inhibitory concentrations (Botić et al., 2017).

 In the text we added the following details:

Discorhabdin G was first isolated in 1995 from an Antarctic collection of Latrunculia apicalis, reported as antibacterial and active in causing a feeding deterrence behavior on one of the major Antarctic sponge predators [12]. The cytotoxicity of this compound is considerably lower than other discorhabdins, as reported for a large library of discorhab-din alkaloids (including discorhabdin G) assessed for effects on Merkel cell carcinoma viability which revealed no apparent mechanistic differences between different dis-corhabdin metabolites tested. The results of this study suggest that these compounds do not induce apoptosis, but rather induce mitochondrial dysfunction leading to non-apoptotic cell death [11].

3. The assertion that removing the spirodienone moiety might diminish cytotoxicity needs substantiation. It would be pertinent to include EC50 data for compound 3 to validate this claim.

From the authors: We fully agree that the assessment of the cytotoxic potential of the compound 3, that showed promising potential as a drug lead in our study, would be very appropriate. However, our experimental approach was mainly devoted to the assessment of inhibitory potential of the new compounds on three different enzymes and as a consequence, we are lacking the compound 3. However, in the near future we plan to test its cytotoxicity on normal and several transformed cell lines, and compare it with other discorhabdin alkaloids. Also, we plan testing its possible undesirable effects on neurotransmission and skeletal muscle.

We have emphasised this also in the conclusions:

“Not the least, the results obtained for compound 3 are relevant to support the efficacy of the approach adopted and serve as a starting point for lead optimization after assessing its possible cytotoxicity, as well as its putative undesirable effects on neuromuscular transmission and skeletal muscle.”

4. Comprehensive kinetic studies comparing the interactions of compound 3 with AChE and BChE would enrich the findings, offering a clearer comparison of its efficacy against these enzymes.

From the authors: An additional figure (Figure S3) representing the inhibition of the three tested cholinesterases by different concentrations of the compounds 3, 7, and 8 has been included to the Supplementary file and cited in the main text. These graphs clearly demonstrate the highest inhibitory potential of the compound 3 over the compounds 7 and 8, as well as its selectivity for AChEs over BChE.

Round 2

Reviewer 2 Report

Comments and Suggestions for Authors

The revised version of the manuscript has successfully addressed the reviewers' comments, looks good.